

# Early identification of recurrence in ovarian cancer: a comparison between the ovarian cancer metastasis index and CA-125 levels

Fei Wang[*], Xuejun Zhao[*], Wenhua Tan, Wei Liu, Yuxia Jin and Qian Liu

Department of Gynecology and Obstetrics, Second Affiliated Hospital of Harbin Medical University, Harbin, China

[*] These authors contributed equally to this work.

## ABSTRACT

Ovarian cancer (OC) is the second most common gynecologic malignancy. A clinical observational study was performed to investigate whether indicators that assess the risk of metastasis can identify recurrence earlier in OC patients. By successfully recruiting 41 patients with OC who underwent chemotherapy, we compared cancer antigen-125 (CA-125) and the ovarian cancer metastasis index (OCMI), which was previously developed by us in the clinic for this purpose. Our results showed that patients and their families generally took a sensible attitude toward disease progression and were willing to accept a new way to gain knowledge about the disease. Herein, the new way was the possibility of monitoring recurrence by introducing the OCMI into the clinic. Fifteen patients experienced recurrence during chemotherapy, implying treatment failure. For 53% of these patients, an abnormally high OCMI suggested a strong tendency toward metastasis at least one chemotherapy cycle prior to the pathological examination confirming recurrence. In comparison, the early recognition rate of recurrence using CA-125 levels was merely 13%. Furthermore, we found that the mean values of the OCMI no longer declined after the fourth chemotherapy cycle, implying that excessive chemotherapy brings no benefit to OC patients. In conclusion, our findings provide a novel and feasible approach to monitor the effectiveness of chemotherapy in the treatment of OC by assessing the potential risk of metastasis.

Corresponding author
Qian Liu, liuqian@hrbmu.edu.cn,
luoluo_liu@sina.com

## INTRODUCTION

Large scale epidemiological studies have shown that ovarian cancer (OC) is the second most common gynecologic malignancy, after breast cancer (*Brawley, 2015*; *Torre et al., 2015*; *Siegel, Miller & Jemal, 2017*). Worldwide, the number of newly diagnosed OC cases was close to 220,000 in 2008, but this number increased to about 240,000 in 2012 (*Jemal et al., 2011*; *Ferlay et al., 2015*). It is expected that this number will increase further due to population aging. Despite having a lower incidence rate than breast cancer, OC causes the largest number of gynecological cancer-associated deaths (more than 100,000 deaths per year). One important reason is that it is difficult to detect early stage OC in a timely

fashion due to the lack of effective biomarkers for early screening (*Menon, Griffin & Gentry-Maharaj, 2014*). The relatively low incidence of OC inevitably increases the relative cost of early screening procedures and the potential risk of erroneous identification (*Henderson, Webber & Sawaya, 2018*). This raises the threshold for the screening of rational biomarkers.

Undoubtedly, the discovery of clinically available biomarkers for early screening will greatly reduce the mortality rate of OC. Before this discovery becomes a reality, an easier and more feasible strategy is to provide more valuable treatment management for patients who have been diagnosed with OC. Surgery plus chemotherapy is the present mainstream OC therapeutic strategy. As about 70% of OC cases are diagnosed at advanced stages (*Jayson et al., 2014*), multi-cycle chemotherapy is unavoidable. Therefore, a big challenge is how to avoid treatment failure due to drug resistance, which is a common occurrence. In the clinic, about half of OC patients face this unfortunate outcome, undergo recurrence, and eventually succumb to the disease (*Marcus et al., 2014*; *Au et al., 2015*). One solution is to develop prognostic biomarkers that can detect recurrence as early as possible. A successful early warning provides doctors an invaluable opportunity to redesign the treatment plan. Up to now, only two biomarkers for early detection of OC recurrence have been approved by the US Food and Drug Administration: cancer antigen-125 (CA-125) and human epididymis protein 4 (HE4). However, the effectiveness of any individual biomarker or a combination of the two has been shown to be very limited (*Van Gorp et al., 2011*).

According to the official definition of the International Federation of Gynecology and Obstetrics, OC can be divided into four stages, stage I to stage IV (*Prat & FIGO Committee on Gynecologic Oncology, 2014*). Stage I indicates no tumor cell migration event has occurred. However, the next three stages indicate an increasing range of tumor cell dissemination from the ovaries to other organs. Multi-organ metastasis often indicates a more serious disease status and lower chance of survival (*Bast Jr, Hennessy & Mills, 2009*). Based on the inner logic of this definition, we developed a novel indicator, the ovarian cancer metastasis index (OCMI), which integrates CA-125 levels with six routine clinical examination indicators by a neural network cascade (NNC)-multiple linear regression hybrid model (*Qu et al., 2018*). NNC is an artificial neural network (ANN) with a serial data input architecture, in which multiple small ANN units that handle single input parameter are connected in tandem to complete the overall prediction task (*Li et al., 2015*; *Hou et al., 2016*; *Cui et al., 2018*). Our work suggested that the OCMI could successfully identify the existence and extent of multi-organ metastasis for a given patient with OC.

Metastasis suggests the existence of active cancer cells. This may further imply resistance to chemotherapeutic agents and the possibility of recurrence. Based on this reasonable hypothesis, we performed this clinical study, in which 41 patients with OC were recruited. In this study, we investigated the potential logical relationship between abnormal OCMI values and the recurrence event that is going to happen during the next period of multi-cycle chemotherapy. Such an effort was aimed to establish an association between metastasis and recurrence in OC and develop a novel biomarker for early prediction of recurrence.

## MATERIALS AND METHODS

### Ethical statement

This work was a clinical observational study that was approved by the Ethics Committee of the Second Affiliated Hospital of Harbin Medical University (Approval number: KY2017-217) and carried out in accordance with the Declaration of Helsinki. Each participant recruited in this study was informed of the project, signed a written consent form, and completed a short questionnaire (File S1). The purpose of the questionnaire survey was to identify patients' or family members' attitude about early prediction of recurrence. The family members and the patients were considered to have the same decision-making ability. In the entire study, no biological samples were collected from any patient.

### Inclusion and exclusion criteria

All the patients enrolled in this study were diagnosed with epithelial ovarian cancer and underwent multi-cycle chemotherapy. Patients were excluded from the study if they were diagnosed with another malignant tumor or endometriosis.

### Patient information collection

For each patient, the information in her medical record was use to confirm recurrence and to calculate the OCMI, following the method established previously (Qu et al., 2018). The information included the patient's age, pharmacotherapeutic regimen, number of metastatic organs confirmed during surgery, ascites, laterality, imageologically-diagnosed or pathologically-confirmed recurrence, and four blood test indicators (CA-125, lymphocyte percentage, prealbumin, and blood platelet count) that were measured at the time of diagnosis, before surgery, and before each cycle of chemotherapy.

### Calculation of the OCMI

OCMI values were calculated for each patient when the four blood test indicators mentioned above were available, typically at the time of diagnosis, before surgery, and before any cycle of chemotherapy. An established neural network cascade (NNC)-multiple linear regression (MLR) hybrid model (Qu et al., 2018) was directly used for this calculation. The hybrid model contained an NNC and an MLR formula. The NNC, an artificial neural network model of a serial data input architecture (Li et al., 2015; Hou et al., 2016; Cui et al., 2018), was previously built using a training dataset containing 534 patients with OC (Qu et al., 2018). Briefly, the four blood test indicators were normalized into a 0 to 1 digital number before further use, as previously described (Zhu & Kan, 2014). After that, the normalized values of the four blood test indicators were inputted into the NNC, and the NNC output was calculated using STATISTICA Neural Networks (SNN, Release 4.0E; Statsoft, Tulsa, OK, USA). Finally, the NNC output, ascites, and laterality were inputted into the MLR formula to calculate the OCMI value. If no ascites was found, the variable was valued at 0; otherwise, it was valued at 1. For the laterality variable, a value of 0 represented unilateral OC and a value of 1 indicated bilateral OC.

### Determination and validation of metastasis discrimination thresholds

The training dataset containing 534 patients with OC (*Qu et al., 2018*) was used to determine thresholds for CA-125 levels, the NNC, and the OCMI for discriminating whether metastasis occurs in a patient with OC (Table S1). MedCalc version 15.8 (MedCalc, Mariakerke, Belgium) was used, and receiver operating characteristic (ROC) curve analysis was performed to obtain the optimal thresholds of CA-125 levels, the NNC, and the OCMI. In addition to the area under the ROC curve (AUROC), the sensitivity, specificity, accuracy, and Youden index were calculated at the optimal cut-off point. The Youden index was the sum of the sensitivity and the specificity minus 1, as defined previously (*Youden, 1950*). Additionally, an independent validation set of 267 patients with OC (*Qu et al., 2018*) was used to validate the prediction accuracies of metastasis identification using the optimal thresholds of CA-125 levels, the NNC, and the OCMI (Table S2).

### Statistical analyses

All data are expressed as mean ± standard deviation. Spearman's correlation test and Chi square test were performed using Graphpad Prism version 6.0 (GraphPad Software, Inc., La Jolla, CA, USA). MedCalc version 15.8 was applied to perform ROC curve pairwise comparison based on the methodology of *DeLong, DeLong & Clarke-Pearson (1988)*. Differences were only considered to be significant at $p < 0.001$.

## RESULTS

### The questionnaire showed good acceptability of the OCMI from the patient aspect

In our previous study (*Qu et al., 2018*), metastasis was defined that metastasis in any one of the 15 organs (bladder, diaphragm, greater omentum, internal genital organ, large intestine, liver, lymph node, mesentery, paracolic sulci, peritoneum, rectouterine fossa, small intestine, spleen, stomach, and ureter) was intraoperatively examined for a patient. And multi-organ metastasis means that more than one organs were found to have cancer metastasis in the intraoperative examination. Based on the good ability of the OCMI for identifying multi-organ metastasis in OC patients (*Qu et al., 2018*), we hoped to introduce the OCMI into clinical practice. This work is an exploratory study for this purpose. A questionnaire was designed and applied to investigate whether patients and patients' families can accept the OCMI-based early identification of recurrence (File S1). A total of 22 patients and 28 patients' families were invited to participate in the questionnaire (Fig. 1A). The response rate was 82%. Only six patients and three patients' families refused to fill out the questionnaire. Finally, the necessary medical record information of 41 patients was collected to calculate the OCMI (Table S1).

Although 16 patients and 25 patients' families agreed to participate in the study, more than two-thirds of the participants expressed doubts about the OCMI (Fig. 1B). However, most of the responders expressed the desire to receive the OCMI test results in the shortest possible time (Fig. 1C). Six patients chose to face the result alone, but the majority of patients chose to consult with their families, regardless of the result (Fig. 1D). Finally, all

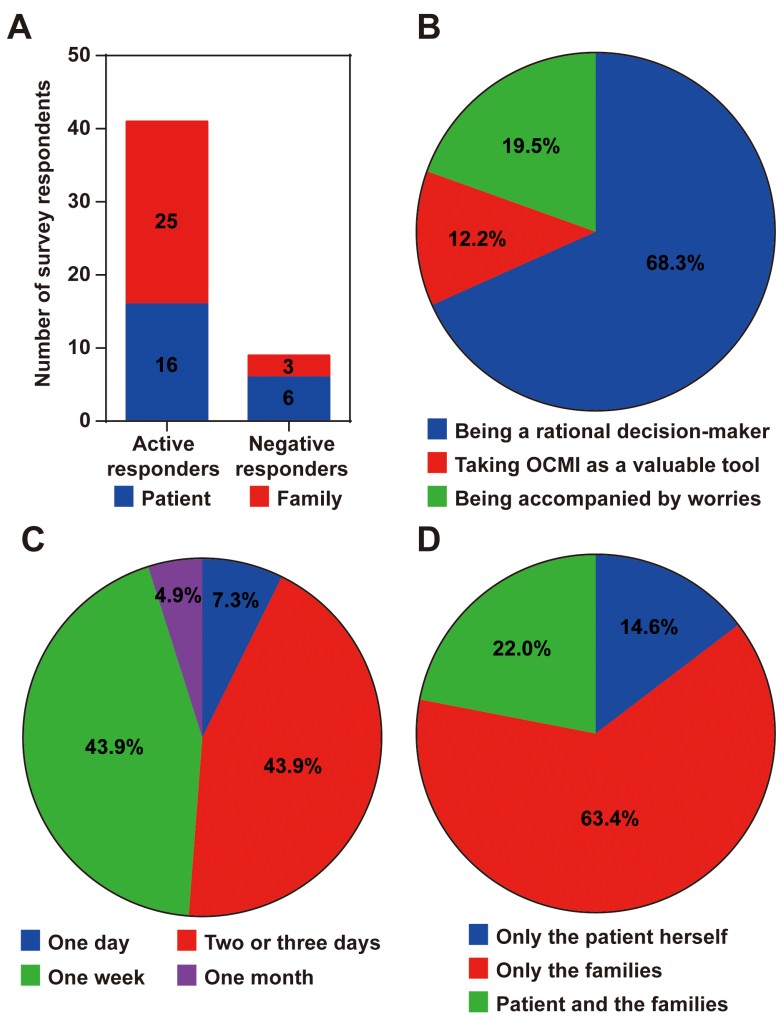

**Figure 1 Results of the questionnaire.** (A) Distribution of questionnaire respondents. (B) Respondents' attitudes toward the OCMI. (C) The time-cost of the OCMI result. (D) Persons that were informed of the OCMI result. OCMI, ovarian cancer metastasis index.

the participants agreed to adopt the OCMI for disease management during hospitalization and after discharge.

## The OCMI has a stronger metastasis risk recognition ability than CA-125 levels

A published dataset of 534 patients with OC (*Qu et al., 2018*) was used to set the metastasis discrimination thresholds of the OCMI and CA-125 levels (Table S2). Our previous study had shown a greater advantage of the OCMI in identifying multi-organ metastasis than the NNC or CA-125 levels (*Qu et al., 2018*). When using the optimal cut-off points, the OCMI had a higher accuracy in multi-organ metastasis risk recognition than CA-125 levels (Table 1). In the present study, ROC curve comparison analysis was performed to investigate the comparative advantage of the OCMI in identifying potential metastasis risk

**Table 1  Receiver operating characteristic curve analysis results of identifying multi-organ metastasis.**

| Index | Threshold | Sensitivity (%) | Specificity (%) | Youden index J | Accuracy (%) |
|-------|-----------|-----------------|-----------------|----------------|--------------|
| CA-125 | 376 | 80.6 | 52.3 | 0.329 | 66.5 |
| NNC | 0.565 | 71.3 | 69.9 | 0.412 | 70.6 |
| OCMI | 0.558 | 84.0 | 68.8 | 0.528 | 76.4 |

(Fig. 2). The OCMI successfully identified metastasis better than the NNC or CA-125 levels ($p < 0.001$). The AUROC for the OCMI was 0.856, but that for CA-125 levels was 0.752. When using the optimal cut-off point, the prediction accuracy of the OCMI was 77.2%, which was superior to that of CA-125 levels (74.0%; Table 2).

### The OCMI achieves early identification of recurrence better than CA-125 levels

For each patient, blood test indicators (CA-125, lymphocyte percentage, prealbumin, and blood platelet count) were recorded before each cycle of chemotherapy, and the OCMI was calculated using the method previously described (Qu et al., 2018). An obvious decline in the CA-125 levels was observed from the beginning of chemotherapy to the fifth chemotherapy cycle. However, after that time, the CA-125 levels no longer decreased (Fig. 3A). A similar trend was observed with the OCMI (Fig. 3B). During our observation period, 15 patients experienced recurrence (Table S1). It should be noted that 60% of the recurrence events happened at the seventh chemotherapy cycle, whereas no recurrence was observed from the beginning of chemotherapy to the fifth chemotherapy cycle. By using multi-organ metastasis discrimination thresholds (CA-125 = 376 U/mL and OCMI = 0.558), two patients had early identification of recurrence by CA-125 levels, but 6 had early identification using the OCMI. Following the metastasis discrimination thresholds (CA-125 = 313 U/mL and OCMI = 0.504), two patients who underwent recurrence were identified early by CA-125 levels, but eight were identified early by the OCMI (Fig. 3C). On the contrary, among the 26 non-recrudescent patients, only three had OCMI values higher than 0.504 from the beginning of chemotherapy to the fifth chemotherapy cycle. Comparatively, four non-recrudescent patients had CA-125 levels higher than 313 U/mL during the same period.

## DISCUSSION

Due to the lack of effective early monitoring tools for OC recurrence, we can only take remedial measures after patients develop recurrence (Davidson & Tropé, 2014; Au et al., 2015; Nezhat et al., 2015; Bowtell et al., 2015). In this exploratory work, we attempted to establish a link between metastasis and recurrence using a quantitative index that was developed to assess the degree of metastasis in OC patients (Qu et al., 2018). The OCMI is an integrated index generated from six conventional clinical examination indicators. One advantage of not developing a separate and new clinical test is that it can be immediately brought into clinical practice. According to the definition of the index, a greater OCMI indicates a wider range of metastasis or a higher metastasis risk. In the present study, we

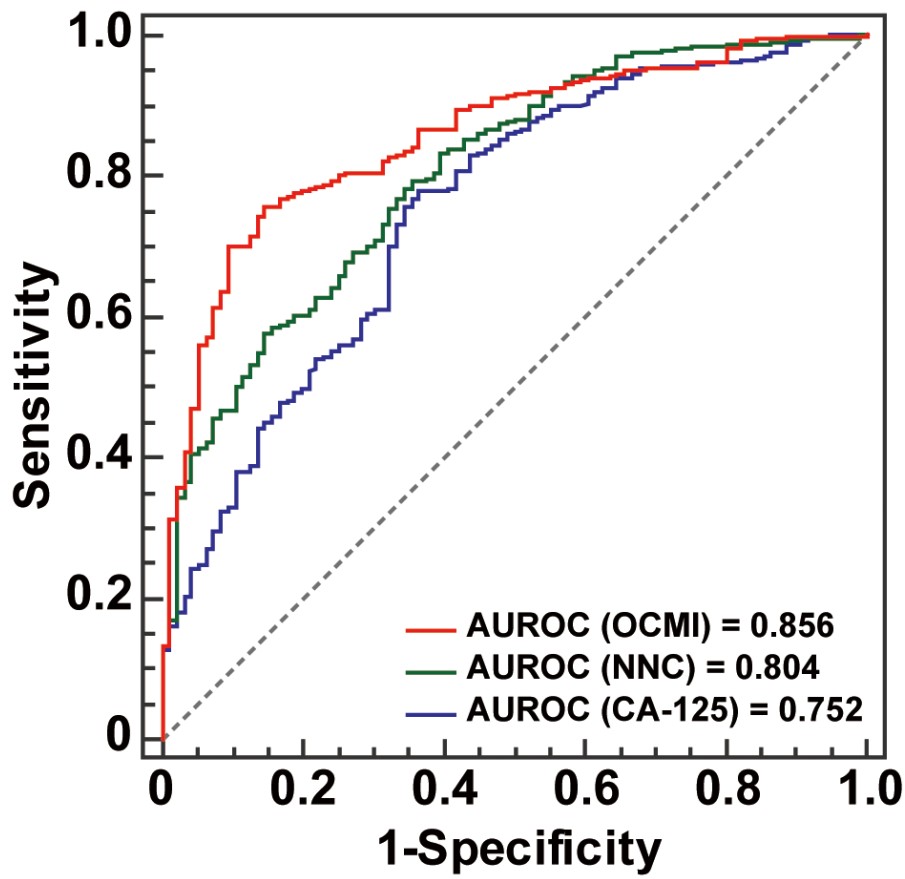

**Figure 2  AUROC comparison results.** NNC, neural network cascade; OCMI, ovarian cancer metastasis index; AUROC, area under the receiver operating characteristic curve.

further confirmed that the OCMI could identify recurrence early by identifying patients at high risk of metastasis. It has been generally accepted that metastasis is the main indicator of the pathological development of OC (*Bast Jr, Hennessy & Mills, 2009*; *Prat & FIGO Committee on Gynecologic Oncology, 2014*). This is also the logical basis for this study. Several limitations of this study should be taken into consideration. One limitation is that the sample size of this single center study is small, and therefore, the conclusions made here need to be re-validated in a larger multicenter study in the future. Another limitation is that the early recognition rate of recurrence by the OCMI is still not satisfactory, and further improvement of the OCMI is needed in the future. One feasible direction is to integrate more useful indicators into the calculation of the OCMI.

Suffering from a malignant tumor causes patients to experience a greater depth of loneliness, but does not generally cause them to be overly pessimistic (*Çıracı, Nural & Saltürk, 2016*; *Thieme et al., 2017*). Consistently, our findings suggest that the vast majority of OC patients and their families could face the disease rationally when a new possibility appeared. They chose to accept it tentatively instead of rejecting it. Active participation of the vast majority of OC patients and their families implies an embodiment of optimism

**Table 2  Receiver operating characteristic curve analysis results of identifying metastasis.**

| Index | Threshold | Sensitivity (%) | Specificity (%) | Youden index J | Accuracy (%) |
|-------|-----------|-----------------|-----------------|----------------|--------------|
| CA-125 | 313 | 75.8 | 65.6 | 0.414 | 74.0 |
| NNC | 0.458 | 78.3 | 65.6 | 0.439 | 75.8 |
| OCMI | 0.504 | 75.6 | 85.4 | 0.607 | 77.2 |

rather than passive acceptance due to excessive pessimism. Although our method was not able to provide a completely new and promising treatment, patients still wanted to know their OCMI test results as early as possible. This tendency indicates that good treatment and deep understanding of the disease are equally important to OC patients. Undoubtedly, valuable information can help them make decisions that are more in line with their interests. Throughout the study, we were always aware of patients' desire for valuable information about their disease condition.

On the one hand, the lack of necessary disease assessment information often leads doctors to tentatively implement overtreatment; on the other hand, patients' psychological acceptance is often underestimated by doctors and therefore it is difficult for patients to get information that is necessary for disease self-assessment. This constitutes a vicious cycle. Overtreatment will undoubtedly maximize patients' harm and minimize the benefit of clinical treatment for cancer (*Loeb et al., 2014*; *Mukhtar, Wong & Esserman, 2015*). When a malignant disease, such as OC, has a high probability of treatment failure, disease assessment information is more meaningful for patients. This helps them have more opportunities to rationally design their remaining time and make it more valuable.

In our study, we considered two threshold standards for the OCMI. A high threshold (OCMI = 0.558) was produced from a previous ROC curve analysis for identifying multi-organ metastasis. Our result indicates that only 40% of the patients with recurrence could be identified in advance of five chemotherapy cycles. By using the MedCalc software, a lower threshold (OCMI = 0.504) was generated in this study from a ROC curve analysis for identifying metastasis, rather than multi-organ metastasis. Our finding indicates that application of the lower threshold of OCMI led to a higher recognition rate of patients with recurrence. Eight of the 15 patients with recurrence were identified in advance of at least one chemotherapy cycle. Meanwhile, only three of the 26 patients without recurrence were suggested to have a high risk of relapse. Comparatively, use of CA-125 alone, a specific biomarker of OC, failed to identify patients with recurrence before the recurrence was confirmed by clinicopathological examination. The early recognition rate of recurrence was 13% using CA-125 levels as a biomarker. Furthermore, our findings also suggest that increasing the number of chemotherapy cycles did not further reduce the risk of cancer metastasis in OC patients. Clinical observations demonstrated that patients could only show a limited response to over-chemotherapy (*Bowtell et al., 2015*) and over-chemotherapy led to a greater possibility of recurrence (*Agarwal & Kaye, 2003*). Our findings further suggest that over-chemotherapy does not provide more benefits in blocking cancer metastasis than moderate chemotherapy does.

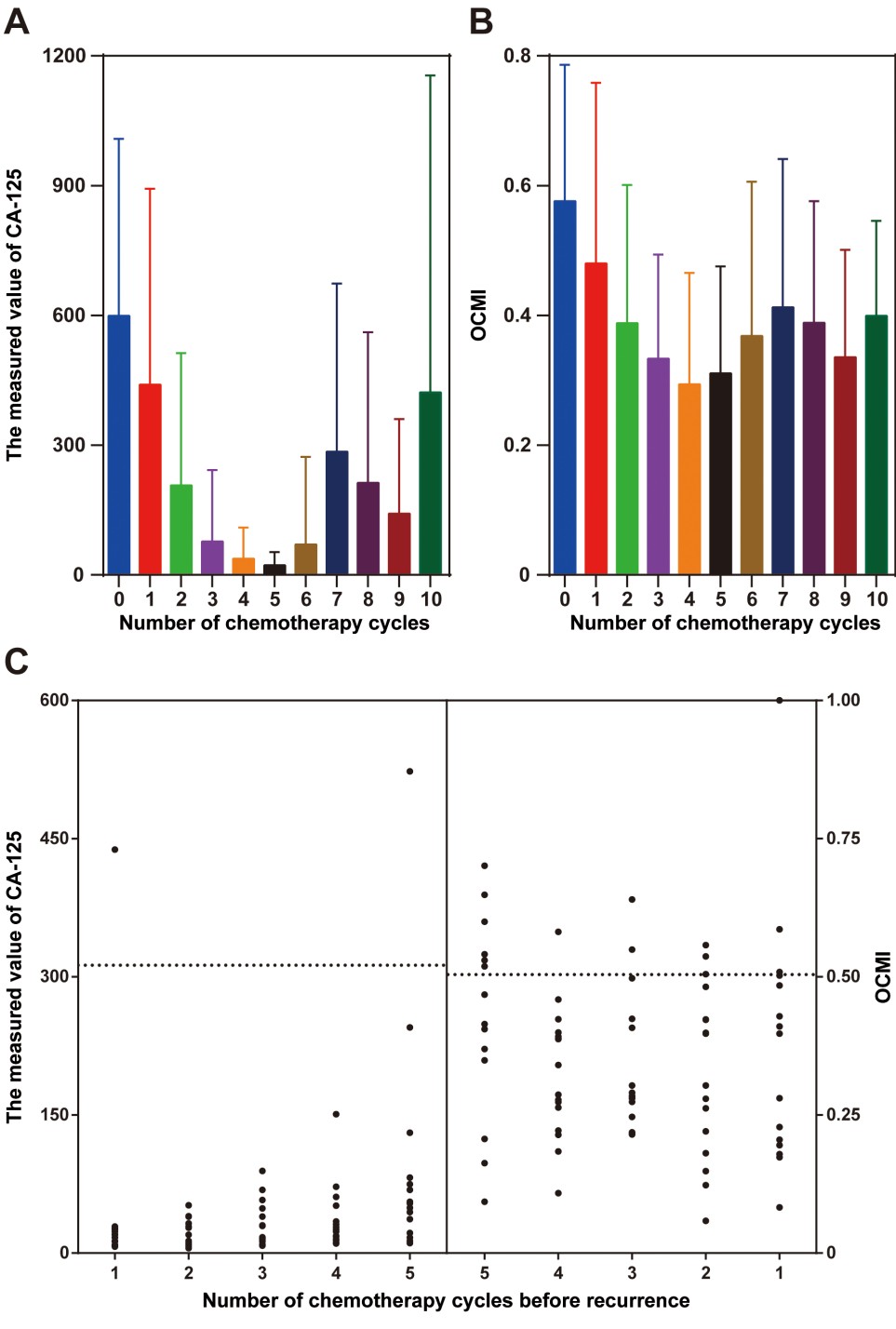

**Figure 3** **Comparison between CA-125 levels and the OCMI in the early identification of recurrence.**
Variation tendencies of CA-125 levels (A) and the OCMI (B). (C) Scatter plots of CA-125 values and
OCMI values in recurrent patients (five chemotherapy cycles before recurrence). The left dotted line
represents the CA-125 metastasis threshold (CA-125 = 313 U/mL) and the right dotted line indicates the
metastasis threshold using the OCMI (OCMI = 0.504). OCMI, ovarian cancer metastasis index; CA-125,
cancer antigen-125.

In conclusion, we provide a new and useful tool for the early identification of relapse during chemotherapy administration in OC patients. Our findings suggest that model refining of only a few of routine clinical indicators can bring a new possibility of early prediction of cancer recurrence. More importantly, this method means a valuable opportunity to replace the failed chemotherapy regimen before recurrence really occurs. We believe that introducing the OCMI into clinical practice represents a feasible and low-cost strategy and the OCMI can fill in the information shortage of clinical disease management of OC, which is caused by low efficiency of CA-125.

### Funding

This work was supported by the National Natural Science Foundation of China (No. 81401502), the Natural Science Foundation of Heilongjiang Province (No. H2018020) and the Post-Doctoral Foundation of Heilongjiang Province (No. LBH-Q17120). The funders had no role in study design, data collection and analysis, decision to publish, or preparation of the manuscript.

### Grant Disclosures

The following grant information was disclosed by the authors:
National Natural Science Foundation of China: 81401502.
Natural Science Foundation of Heilongjiang Province: H2018020.
Post-Doctoral Foundation of Heilongjiang Province: LBH-Q17120.

### Competing Interests

The authors declare there are no competing interests.

### Author Contributions

- Fei Wang and Xuejun Zhao performed the experiments, analyzed the data, prepared figures and/or tables, authored or reviewed drafts of the paper.
- Wenhua Tan, Wei Liu and Yuxia Jin performed the experiments.
- Qian Liu conceived and designed the experiments, contributed reagents/materials/analysis tools, approved the final draft.

### Human Ethics

The following information was supplied relating to ethical approvals (i.e., approving body and any reference numbers):

This work was a clinical observational study that was approved by the Ethics Committee of the Second Affiliated Hospital of Harbin Medical University (Approval number: KY2017-217).

### Data Availability

The raw measurements are available in Tables S1 and S2.

# PeerJ

## Supplemental Information

Supplemental information for this article can be found online at http://dx.doi.org/10.7717/peerj.5912#supplemental-information.

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
