# Peer review of "Early identification of recurrence in ovarian cancer: a comparison between the ovarian cancer metastasis index and CA-125 levels"

_PeerJ, doi:10.7717/peerj.5912_

## Round 0.1 · original submission · Major Revisions

Dear Dr Qian,

Please address all the Reviewers concerns including those in the annotated MS.

Reviewer 1 ·

Basic reporting

In general I believe that it is a quality paper with interesting and relevant results, taking into account the small sample size as mentioned by the authors.
The English would require some small improvements (please refere to the comments I the attached file).
Literature references are adequate.

Experimental design

This paper deals with an important and original topic, supported by robust methods and statistics. Moreover it meets the aims and scope of this journal.
The authors adequately included the informed consent.

Validity of the findings

As above mentioned this paper addresses an important and novel subject showing interesting and relevant results. The presented statistics are meticulous. However, the results need to be further validated with larger samples.
In my opinion, the conclusion could be rewritten in a more concisely and straightforward manner.

Additional comments

No further comments

Annotated reviews are not available for download in order to protect the identity of reviewers who chose to remain anonymous.

Reviewer 2 ·

Basic reporting

The English language should be carefully revised to improve the clarity of the scientific message. For example, lines 60 (“Thus far, …”), line 75 (”By such an …) and line 191-193 (“The high degree…”) should be rephrased.

Experimental design

In the introduction, the authors did not refer to NCC which is used to compare results with CA-125 assay in the result section (Table 1, Table 2 and Figure 2). OCMI should be better defined in the introduction as a hybrid model of NNC and multiple linear regression as defined in the Material and Methods section.

Results Section: Metastasis and multi-organ metastasis should be better defined. Metastasis refer to peritoneal metastasis? And Multi-organ metastasis refer distant metastasis (e.g. lung) or refer to organs within the peritoneal cavity (e.g.liver)

Figure 1: Graphic A should be reformulated to show that within the “Active responders” which ones respond negative.

Validity of the findings

Discussion: The final message should be positive and demonstrative of the strong points of the presented work. The limitations should be discussed in the beginning of the discussion.

Additional comments

Wang F. et al presented a work with ovarian cancer patients to evaluate the precision rate of the ovarian cancer metastasis index (OCMI) to predict disease recurrence. The results where compared with classical follow up biomarker (CA-125) and Neural network cascade (NCC). Authors showed that OCMI ROC was 0.856, NNC ROC 0.804 and CA-125 alone was 0.752. Moreover, OCMI level predicted recurrence earlier than CA-125 values (1 chemotherapy cycle earlier). The improvement of OC recurrence prediction is of paramount importance and the idea of refining the already available follow up methods would accelerate the implementation of this improved method in to the clinical. The aim of this work is therefore very significant but the present manuscript require some alteration in order to clarify the results.

---

## Round 0.2 · Minor Revisions

Dear Dr Wang,

Please do the remaining minor modifications suggested by Reviewer 1 in the attached file.

Reviewer 1 ·

Basic reporting

In general I believe that it is a quality paper with interesting and relevant results, taking into account the small sample size as mentioned by the authors.
The English would still require some small improvements (please refere to the comments I the attached file).
Literature references are adequate.

Experimental design

This paper deals with an important and original topic, supported by robust methods and statistics. Moreover it meets the aims and scope of this journal.
The authors adequately included the informed consent.

Validity of the findings

As above mentioned this paper addresses an important and novel subject showing interesting and relevant results. The presented statistics are meticulous. However, the results need to be further validated with larger samples.

Additional comments

No further comments

Annotated reviews are not available for download in order to protect the identity of reviewers who chose to remain anonymous.

Reviewer 2 ·

Basic reporting

As suggested the English language was improved.

Experimental design

The authors addressed all questions and/or suggestions required to refine the scientific message of the manuscript.

Validity of the findings

As suggested the Discussion section was improved.

Additional comments

The authors addressed all questions and/or suggestions required to publication.

---

## Round 0.3 · accepted · Accept

The authors have addressed the minor comments requested